# Effects of Elevated $CO_2$ on Wheat Yield: Non-Linear Response and Relation to Site Productivity

**Malin C. Broberg [1],\***, **Petra Högy [2]**, **Zhaozhong Feng [3]** and **Håkan Pleijel [1]**

1 Department of Biological and Environmental Sciences, University of Gothenburg, P.O. Box 461, SE-40530 Göteborg, Sweden; hakan.pleijel@bioenv.gu.se
2 Institute of Landscape and Plant Ecology, University of Hohenheim, Ökologiezentrum 2, August-von-Hartmann Str. 3, D-70599 Stuttgart, Germany; petra.hoegy@uni-hohenheim.de
3 Collaborative Innovation Center of Atmospheric Environment and Equipment Technology, Nanjing University of Information Science & Technology, Nanjing 210044, China; zhaozhong.feng@nuist.edu.cn
\* Correspondence: malin.broberg@bioenv.gu.se; Tel.: +46-701719190

**Abstract:** Elevated carbon dioxide ($eCO_2$) is well known to stimulate plant photosynthesis and growth. Elevated carbon dioxide's effects on crop yields are of particular interest due to concerns for future food security. We compiled experimental data where field-grown wheat (*Triticum aestivum* Linnaeus) was exposed to different $CO_2$ concentrations. Yield and yield components were analyzed by meta-analysis to estimate average effects, and response functions derived to assess effect size in relation to $CO_2$ concentration. Grain yield increased by 26% under $eCO_2$ (average ambient concentration of 372 ppm and elevated 605 ppm), mainly due to the increase in grain number. The response function for grain yield with $CO_2$ concentration strongly suggests a non-linear response, where yield stimulation levels off at ~600 ppm. This was supported by the meta-analysis, which did not indicate any significant difference in yield stimulation in wheat grown at 456–600 ppm compared to 601–750 ppm. Yield response to $eCO_2$ was independent of fumigation technique and rooting environment, but clearly related to site productivity, where relative $CO_2$ yield stimulation was stronger in low productive systems. The non-linear yield response, saturating at a relatively modest elevation of $CO_2$, was of large importance for crop modelling and assessments of future food production under rising $CO_2$.

**Keywords:** carbon dioxide; exposure system; harvest index; grain mass; grain number; grain yield; meta-analysis; response function; site productivity; specific grain mass; *Triticum aestivum*

## 1. Introduction

Over the past four decades, multiple experiments have been performed to estimate the response of plants to higher concentrations of carbon dioxide ($CO_2$) under field conditions. Agricultural crops have been of particular interest due to the strong concerns for future food security and safety [1] and to explore the potential advantages of the fact that rising $CO_2$ may stimulate plant growth. The Intergovernmental Panel on Climate Change [2] projected that $CO_2$ concentrations are likely to be in the interval 420–1300 ppm (RCP2.6 and RCP8.5, respectively) by the year 2100. Consequently, to assess possible yield stimulations, there is a need to estimate crop responses to elevated $CO_2$ ($eCO_2$) over a range of concentrations, although single experiments mostly used only one or sometimes two levels of $eCO_2$ treatment.

Wheat (*Triticum aestivum* Linnaeus) is one of the most studied crops regarding $eCO_2$ responses, since it is one of the major food crops globally. Plant growth is generally stimulated by $eCO_2$, resulting in higher yields [3]. The growth stimulation is a result of both enhanced photosynthesis (C3 crops), but also improved water use efficiency (C3 and C4 crops) due to reduced stomatal conductance [4]. Short-term

plant responses to $eCO_2$ usually include a higher net $CO_2$ assimilation, while downregulation of photosynthesis can occur over longer time scales (growing season) [5]. The $CO_2$ fertilization effect on C3 photosynthesis will mainly occur until the concentration is saturated at ribulose-1,5-bisphosphate carboxylase/oxygenase (Rubisco), which is not the case at the current atmospheric concentration (400 ppm). In addition, the water saving effect can significantly improve plant performance [6,7]. However, it remains uncertain how these effects translate into crop yield responses over a wider range of $CO_2$ concentrations under field conditions.

Enclosure systems, such as open top chambers (OTC), have been widely used in $CO_2$ field experiments, but also questioned since they alter the micro-climate of the plants and thus may modify the magnitude of crop responses to $eCO_2$ [8]. Comparison of conditions in OTCs to the open field show that temperatures and vapor pressure deficits are higher inside chambers and airflow is altered in the plant canopy [9,10]. The use of OTCs will also reduce transmission of solar radiation and shift the ratio between diffuse and total radiation. The field tunnels (e.g., Rawson [11]) used in some $eCO_2$ experiments with crops are likely to alter the micro-climate in a similar manner as OTCs. Furthermore, it is questionable whether results from experiments with plants rooted in pots can be extrapolated to field conditions. Potted plants have a restricted rooting volume that may affect the response to $eCO_2$ [12]. At the same time, plants grown in pots are likely to experience a higher, light interception, since they are usually not surrounded by a closed canopy, which may exaggerate effects. Free Air $CO_2$ Enrichment (FACE) systems have been developed to create a less artificial experimental setup compared to enclosure systems like OTCs and tunnels. On the other hand, FACE systems have the drawback of not being able to reach strongly elevated concentrations for $eCO_2$ treatments (no experiments using $CO_2$ concentrations above 600 ppm) and possibly less stable concentration levels that may lead to underestimation of plant $CO_2$ responses [13].

In addition to grain yield as such, there are a number of yield variables of both agronomical and economical importance for grain yield, which are critical to study in order to understand how $eCO_2$ affects the growth pattern of crops. In the present study we included the following yield components and aspects of grain physical characteristics: harvest index, grain number, grain mass, and specific grain mass. Harvest index represents the fraction of the total aboveground biomass found in the harvestable products at maturity, which is central in crop breeding as a measure of the efficiency with which resources (solar radiation, water, and fertilizers) are used and converted into the desired harvested plant component. Grain mass (equivalent to 1000-grain weight) and specific grain mass (volume weight or test weight) are quality aspects that affect the market price of wheat grain. Higher values of these variables are related to a larger flour yield, while low values indicate small and malformed grains of poor quality [14,15]. Historical improvements in wheat grain yield has been positively correlated to an increase in grain number per unit area [16]. There is, however, a trade-off between increasing the number of grains and grain mass if photosynthetic rates remain unchanged [17]. The $CO_2$-induced grain yield (mass per unit area) stimulation can be a result of increased grain number (per unit area) and/or average grain mass.

The $CO_2$ effect on wheat grain yield was reviewed by Amthor [18], where response functions showed that studies performed in laboratory chambers and greenhouses compared to field experiments had almost doubled yield stimulation per increased ppm in the range of 350–750 ppm. There was, however, only one FACE experiment conducted at that time. Using meta-analysis, Wang, Feng, and Schjoerring [4] estimated the overall $CO_2$ impacts on wheat crop physiology and yield, showing an average yield stimulation of 24%, and an effect of similar magnitude was estimated by van der Kooi et al. [19]. In line with Amthor [18], Wang, Feng and Schjoerring [4] found differences in yield response between enclosure systems, where closed-top chambers had a yield stimulation close to 40%, significantly higher than all other types of exposure systems (OTCs, FACE, and greenhouses). They also found that the grain yield stimulation by $eCO_2$ was significantly stronger in potted plants compared to field grown; however, not taking into account that there is an association between rooting environment and the enclosure system used. Studies with greenhouses and closed-top-chambers

mainly used pots, while plants in FACE experiments were grown in field soil and OTC studies use both potted and field rooted plants.

Our study aims to provide an up-to-date summary of $eCO_2$ effects on wheat crop yield, based on observations from ecologically realistic field experiments and excluding treatments with additional environmental stress, such as heat, drought, elevated ozone, or low nitrogen (N) supply. In addition, we will consider $CO_2$ effects on various wheat yield components (harvest index, total aboveground biomass, grain mass, grain number, and specific grain mass), which was not included by Amthor [18]. There has also been discussions regarding the relationship between crop $CO_2$ response and environmental conditions controlling site productivity [14,20], such as growing season temperature and water availability, which varies considerably between different regions of wheat cultivation. Bishop, Leakey, and Ainsworth [20] concluded that growing season temperature was not a good predictor of yield $CO_2$ response but found a negative relationship between water supply (precipitation + irrigation) and yield $CO_2$ response; however, the dataset was rather limited for this analysis since not many experiments reported total water supply. To address this, we examined the relationship between the $eCO_2$ effect on grain yield and the absolute grain yield in control treatments, used as a proxy for wheat crop productivity in the area where the experiment was performed. In the present study, we used meta-analysis to estimate the overall magnitude and statistical significance of responses to $eCO_2$ treatments. Response functions were derived in order to understand the gradual change over the range of $CO_2$ concentrations used in the experiments, and to test if the response with increasing $CO_2$ was linear or non-linear. By these approaches our study aims to answer the following research questions:

1. Does yield stimulation by $eCO_2$ saturate at high $CO_2$ concentration?
2. Is $eCO_2$ yield stimulation dependent on experimental conditions (fumigation technique, rooting environment, wheat type, geographic region)?
3. Is there a link between $eCO_2$ yield stimulation and agronomic productivity?
4. Are yield components (total aboveground biomass, harvest index, grain mass, grain number, specific grain mass) equally affected by $eCO_2$?

## 2. Materials and Methods

### 2.1. Database

Web of Science, Scopus, and Google Scholar were used to survey peer-reviewed literature published between 1980 and 2017 (October) related to experiments with wheat grown under $eCO_2$ until maturity. Some additional data, not found within these databases but obtained directly from experimenters, were included in the study and references are given in Appendix A. Experimental data were included if any of the following variables were reported: grain yield (g m$^{-2}$/plant$^{-1}$/pot$^{-1}$), harvest index, total aboveground biomass, grain number (m$^{-2}$/plant$^{-1}$/pot$^{-1}$), grain mass (1000-grain weight), and specific grain mass (test weight/volume weight). Experiments performed in greenhouses or closed-top growth chambers were excluded from the analysis in order to avoid the impact of ecologically non-realistic growing conditions. For experiments with a factorial design, treatments with elevated ozone, drought stress, heat stress, or lower than normal fertilizer application rate (relative to local agronomic practice) were excluded from the analysis. References to all data included in the analysis can be found in Appendix A and data are given in the Supplementary Materials File. Data only presented in graphs were extracted using GetData Graphic Digitizer (version 2.26) [21]. In cases where the ambient $CO_2$ concentration ($aCO_2$) was not reported, it was assumed to be equal to the global average for the year the study was conducted, using the Mauna Loa record as the reference [22].

### 2.2. Meta-Analysis

Meta-analysis was performed with MetaWin software package (Version 2.0) [23], using the $eCO_2$ as treatment and $aCO_2$ as control. In line with previous meta-analyses [24,25], observations were considered to be independent if they were made on different cultivars, different $eCO_2$ treatments, and

during different years. The natural logarithm of the response ratio (the ratio of the means of two groups, experimental treatment and control) was used as effect size, and reported as the percentage change from the control. Due to the lack of data for computation of sample variance (standard deviation or standard error with degree of replication), an un-weighted approach was used for the analysis [23,24]. The variance of effect size was calculated using the re-sampling method with 9999 iterations, and confidence intervals (CIs) were calculated with the bootstrap method [26]. The average effect size was considered to be significant if the 95% CI did not overlap zero, and for subgroup analysis the different groups were considered to be significantly different if the 95% CI did not overlap [27]. In the subgroup analysis data was categorized by (1) exposure system: FACE, OTC, and tunnel; (2) rooting environment: plants rooted in pots or field; (3) the concentration level of $eCO_2$ treatment: above or below 600 ppm; (4) region where the experiment was performed: Asia, Australia, Europe, and North America; and (5) wheat type: spring or winter wheat. For the current dataset, the maximum $eCO_2$ level used was 750 ppm, since we have restricted the study to field experiments using FACE, OTC or tunnels.

*2.3. Response Functions*

Response functions were derived through regression between the relative effect on each yield variable and the corresponding $CO_2$ concentration for every observation. Using linear regression the response was related to the effect estimated at 350 ppm for each individual experiment. At 350 ppm the effect variables were set to a value of 0 on a relative scale. Both a first order polynomial (linear) and a second order polynomial (quadratic) model were fitted to the data. Model fit was compared with Akaike information criterion (AICc, including a correction for low sample size), which considers both the difference in residual sum of squares and number of model parameters. Results are presented as a probability of which of the two models that is correct, given by Equation (1) where $\Delta$ = difference in AICc values.

$$probability = \frac{e^{0.5\Delta}}{1 + e^{0.5\Delta}} \tag{1}$$

Response functions were also made for the relationship between the relative treatment effect (response ratio −1%) on grain yield and the absolute grain yield in $aCO_2$ treatments (control). For comparability of absolute grain yield data, only studies reporting grain yield in mass per unit area were included (73 pairs of observations). All response functions were derived using automatic outlier removal [28], outliers also being shown in graphs.

## 3. Results

Figure 1 presents the average effect of $eCO_2$ on a range of wheat yield variables, using $aCO_2$ as the reference. Grain yield significantly increased by 25.6% (CI 20.9–28.5%) under $eCO_2$ and total aboveground biomass showed an equal response of 24.8% (CI 21.7–28.1%). Grain number was significantly enhanced by 22.3% (CI 17.6–27.1%) due to $eCO_2$, while grain mass showed a small, but still significant increase by 2.1% (CI 0.6–3.7%). Harvest index and specific grain mass remained unaffected under $CO_2$ enrichment, effects estimated to 0.4% (CI −1.4–2.2%) and −1.6% (CI −8.8–3.3%), respectively.

The response function for the relationship between grain yield and $CO_2$ concentration (Figure 2a) showed a strong non-linear relationship ($R^2$ = 0.44). Grain yield increased with higher $CO_2$ to reach a maximum yield response at ~600 ppm. Details for all regression models, linear and quadratic, are presented in Table 1. Figure 2b gives the relationship between the grain yield $CO_2$ response (response ratio −1%) with $\Delta CO_2$ (difference in $CO_2$ concentration between $aCO_2$ and $eCO_2$ treatment in each experiment), demonstrating that there was no relationship between grain yield stimulation and difference in concentration between control and elevated treatments.

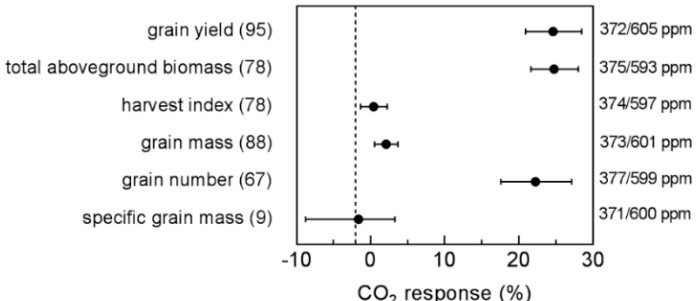

**Figure 1.** Meta-analysis of the $eCO_2$ effect on wheat yield components (grain yield, harvest index, grain mass, grain number, total aboveground biomass, and specific grain mass) using $aCO_2$ as the reference. The number of observation pairs is given within the brackets. The average concentration level for ambient and elevated $CO_2$ treatments is given on the right *y*-axis ($aCO_2/eCO_2$).

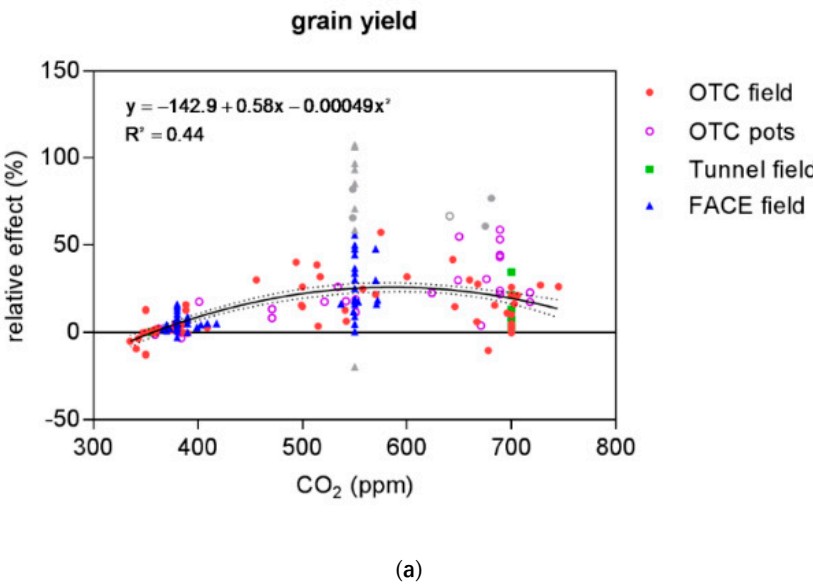

(a)

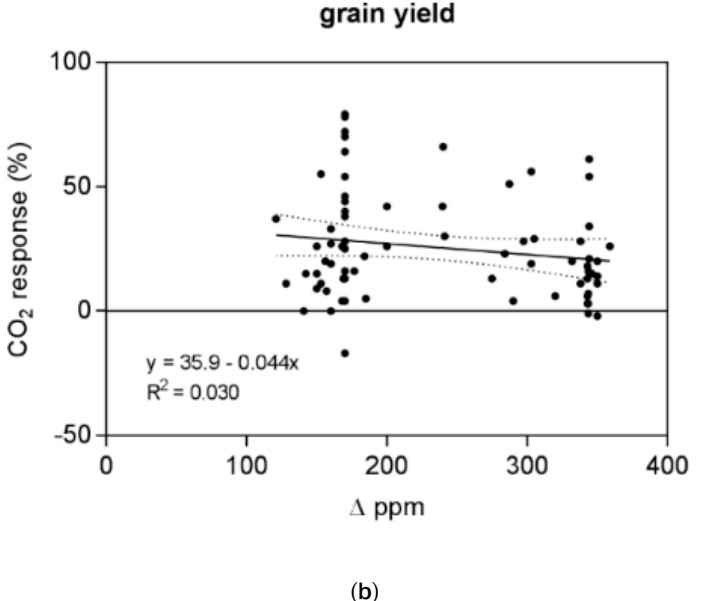

(b)

**Figure 2.** *Cont*.

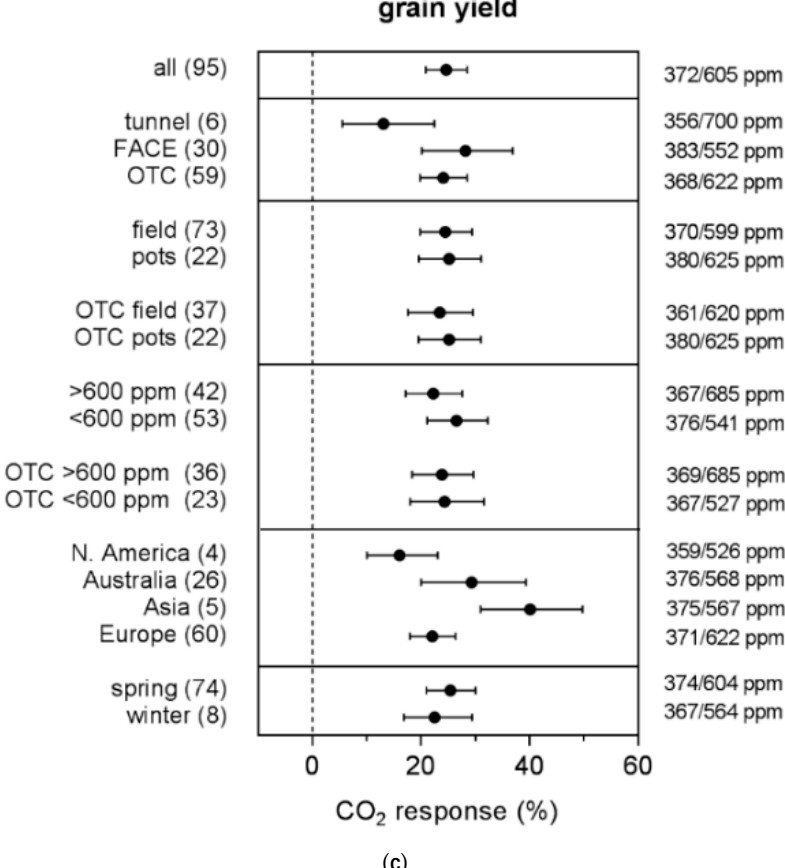

(c)

**Figure 2.** (**a**) Response function for grain yield (relative to yield at 350 ppm) with $CO_2$ concentration. Grey markers are statistical outliers excluded from the curve fitting. (**b**) Relationship between the grain yield $CO_2$ response for and $\Delta CO_2$ (difference in $CO_2$ concentration between $aCO_2$ and $eCO_2$ treatment). Dotted lines represent the 95% confidence interval for regression models. (**c**) Meta-analysis of the $eCO_2$ effect on wheat grain yield using $aCO_2$ as the reference. Subgroup analysis for different exposure systems (FACE, OTC, and tunnel), rooting environment (pots and field soil), concentration level of $eCO_2$ treatment (above or below 600 ppm), region (North America, Asia, Australia, Europe), and wheat type (spring or winter wheat). The number of observation pairs is given within the brackets. The average concentration level for ambient and elevated $CO_2$ treatment is given at the right *y*-axis ($aCO_2/eCO_2$).

**Table 1.** Response functions for regression of yield variables (grain yield, total aboveground biomass, harvest index, grain mass, and grain number) with $CO_2$ concentration [1].

| Variable | Observations (Outliers) | Model | B0 | B1 | B2 | $R^2$ | *p*-Value | Probability % |
|---|---|---|---|---|---|---|---|---|
| Variable | Observations (Outliers) | Model | B0 | B1 | B2 | $R^2$ | *p*-Value | Probability % |
| grain yield | 173 (13) | linear | −17.5 | 0.062 | | 0.29 | <0.0001 | <0.01 |
| | | quadratic | −142.9 | 0.576 | −0.00049 | 0.44 | | >99.99 |
| total aboveground biomass | 139 (4) | linear | −27.1 | 0.086 | | 0.46 | <0.0001 | <0.01 |
| | | quadratic | −123.1 | 0.479 | −0.00038 | 0.54 | | >99.99 |
| harvest index | 139 (10) | linear | 3.2 | −0.007 | | 0.07 | 0.0022 | 49.6 |
| | | quadratic | −5.9 | 0.030 | −0.00004 | 0.09 | | 50.4 |

**Table 1.** *Cont.*

| | | | | | | | | | |
|---|---|---|---|---|---|---|---|---|---|
| grain mass | 159 (0) | linear | 1.4 | 0.001 | | 0.00 | 0.78 | <0.01 |
| | | quadratic | −47.7 | 0.202 | −0.00019 | 0.14 | | >99.99 |
| grain number | 125 (9) | linear | −17.8 | 0.062 | | 0.29 | <0.0001 | <0.01 |
| | | quadratic | −151.9 | 0.061 | −0.00053 | 0.47 | | >99.99 |

[1] Model parameters are presented for both linear ($y = B0 + B1x$) and quadratic ($y = B0 + B1x + B2x^2$) curve fits, $x$ being the $CO_2$ concentration and $y$ being the response variable. The *p*-value is given for the B1 parameter (slope) in linear models. Model fit is compared with the Akaike information criterion (AICc) giving the probability of which of the two models is correct.

Meta-analysis for subgroups of grain yield data (Figure 2c) showed an overall similar response for the different groups. There were no significant differences between exposure systems (FACE, OTC, and tunnels), but an indication of a smaller $eCO_2$ effect in the tunnel systems (13.1.0%, CI 5.5–22.4%) compared to the 24.1% yield stimulation in OTC (CI 19.9–28.5%) and 28.2% in FACE (CI 20.1–36.9%).

There was no significant difference in terms of grain yield between the two levels of $eCO_2$ treatment tested. The average yield response for plants grown in $CO_2$ concentration < 600 ppm was 26.5% (CI 21.1–32.3%) and 22.3% (CI 17.1–27.6%) for concentrations > 600 ppm. The comparison was also made for the subset of OTC data to avoid the potential influence of different types of exposure systems using different levels of $eCO_2$ treatment. Results from OTCs showed no significant differences between $eCO_2$ levels above and below 600 ppm, with an $eCO_2$ effect of 23.9 (CI 18.4–29.7) and 24.4% (CI 18.0–31.6%), respectively. The latter observation (OTC < 600 ppm) was also within the same range as grain yield response in FACE systems, thus when comparing FACE and OTC using the same level of $eCO_2$ there were no significant differences in yield stimulation.

Comparison of the rooting environment showed an almost identical response, where grain yield under $eCO_2$ increased by 25.2% (CI 19.6–31.1%) for potted plants and 24.5% (CI 19.8–29.5%) for plants rooted in field soil. Since potted plants were only used in OTC experiments, this dataset was also compared to a subset of yield data restricted to plants grown in OTC and field soil, where the $eCO_2$ effect was 23.4% (CI 17.6–29.6%), also very similar to the effect for pot grown plants. The average $eCO_2$ concentration in potted plants was 625 ppm, which was comparable to the field grown plants in OTCs with an average of 620 ppm.

Categorizing experiments by region revealed that $eCO_2$ effects on grain yield were significantly stronger in experiments performed in Asia (40.0%, CI 31.0–49.8%) compared to North America (16.0%, CI 10.0–23.1%) and Europe (22.1%, CI 18.0–26.3%), while the Australian experiments (29.3%, CI 20.0–39.3%) did not significantly differ from any of the other regions (North America $p = 0.29$, Europe $p = 0.12$, and Asia $p = 0.39$). It should, however, be noted that the number of observations is small for North America and Asia, with four and five comparisons, respectively. The average $eCO_2$ effect on grain yield in winter wheat was 22.5% (CI 16.9–29.4%), while the response in spring wheat was stronger (25.4%, CI 21.0–30.0%), but with no significant difference between the two wheat types.

Figure 3a shows the relationship between the grain yield $CO_2$ response and absolute grain yield in $aCO_2$, representing the agronomic productivity of the site and year. Data were categorized by exposure system and rooting environment (Figure 3b), concentration level of $eCO_2$ treatment (Figure 3c), and region (Figure 3d). Whatever subdivision of data that was made, the relationship was very similar in all cases, indicating that the estimated average effect of the agronomic productivity on the $eCO_2$ response was robust. Quadratic fit for the complete dataset (Figure 3a) gave a marginally better fit ($R^2 = 0.27$) compared to the linear model ($R^2 = 0.24$) and model performance was equal when considering AICc. Coefficients and model performance for all regression models are presented in Table 2. The relationship between absolute grain yield response and absolute grain yield in $aCO_2$ was also tested but did not show any association ($R^2 = 0.048$).

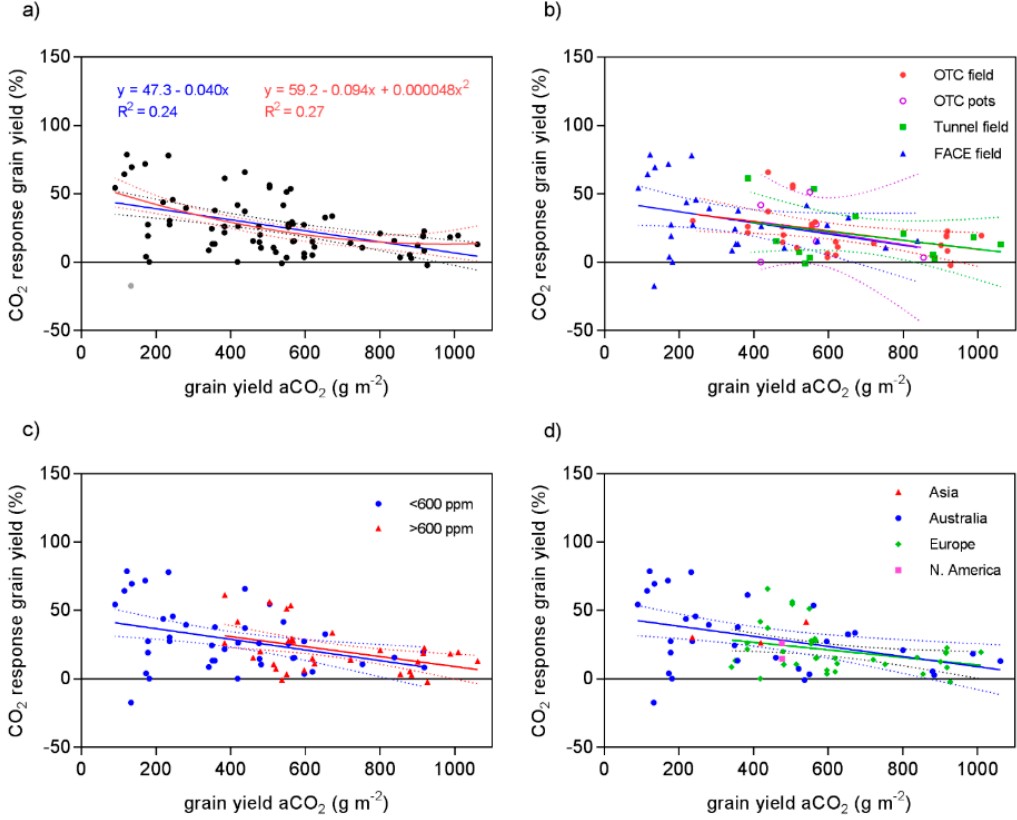

**Figure 3.** Relationship between the relative $CO_2$ response for grain yield and absolute grain yield in $aCO_2$ (control treatment). (**a**) Linear and non-linear regression models for the complete dataset. Subgroups categorized by: (**b**) exposure systems (FACE, OTC, and tunnel) and rooting environment (field or pots), (**c**) concentration level of $eCO_2$ treatment (above or below 600 ppm), and (**d**) region (Asia, Australia, Europe, and North America). Dotted lines represent the 95% confidence interval for the regression models. Grey markers are outliers excluded from the curve fitting. Equations for all regression models are found in Table 2.

**Table 2.** Response functions for the grain yield $CO_2$ response with absolute grain yield in $aCO_2$ (control treatment) [1].

| Group | Observations | Model | B0 | B1 | B2 | $R^2$ | *p*-Value | Probability % |
|---|---|---|---|---|---|---|---|---|
| all | 73 | linear | 47.3 | −0.040 | | 0.24 | 0.00020 | 46.7 |
| | | quadratic | 59.2 | −0.094 | 0.000048 | 0.27 | | 53.4 |
| OTC field | 27 | linear | 43.5 | −0.034 | | 0.19 | 0.022 | 74.9 |
| | | quadratic | 64.1 | −0.10 | 0.000051 | 0.21 | | 25.1 |
| OTC pots | 6 | linear | 47.7 | −0.043 | | 0.11 | 0.52 | >99.99 |
| | | quadratic | −91.3 | 0.42 | −0.00037 | 0.28 | | <0.01 |
| Tunnel field | 12 | linear | 42.0 | −0.032 | | 0.13 | 0.25 | 86.5 |
| | | quadratic | 109.8 | −0.24 | 0.00014 | 0.20 | | 13.5 |
| FACE field | 28 | linear | 45.3 | −0.041 | | 0.11 | 0.079 | 73.9 |
| | | quadratic | 56.4 | −0.11 | 0.000085 | 0.13 | | 26.1 |
| <600 ppm | 42 | linear | 44.6 | −0.039 | | 0.14 | 0.016 | 72.4 |
| | | quadratic | 51.7 | −0.079 | 0.000043 | 0.15 | | 27.6 |
| >600 ppm | 31 | linear | 45.5 | −0.036 | | 0.19 | 0.015 | 56.5 |
| | | quadratic | 102.7 | −0.21 | 0.00012 | 0.24 | | 43.5 |

**Table 2.** *Cont.*

| Group | Observations | Model | B0 | B1 | B2 | $R^2$ | *p*-Value | Probability % |
|---|---|---|---|---|---|---|---|---|
| Australia | 32 | linear | 46.0 | −0.037 | | 0.14 | 0.022 | 76.4 |
| | | quadratic | 51.5 | −0.068 | 0.000030 | 0.17 | | 23.6 |
| Europe | 36 | linear | 37.9 | −0.028 | | 0.11 | 0.050 | 78.1 |
| | | quadratic | 37.2 | −0.026 | −0.0000015 | 0.11 | | 21.9 |

[1] Model parameters are presented for both linear ($y = B0 + B1x$) and quadratic ($y = B0 + B1x + B2x^2$) curve fits, $x$ being grain yield in $aCO_2$ (control treatment) and $y$ being the response ratio –1% for grain yield. The *p*-value is given for the B1 parameter (slope) in linear models. The model fit is compared with the Akaike information criterion (AICc) giving the probability of which of the two models is correct.

The relationship between $CO_2$ concentration and total aboveground biomass (Figure 4a) showed a strong positive non-linear relationship ($R^2 = 0.54$) and the response function for grain number (Figure 4d) had a similar response pattern ($R^2 = 0.47$). Similar to the response function for grain yield, also for total aboveground biomass and grain number, the stimulation by higher $CO_2$ gradually declined and reached a maximum response at ~600 ppm. The response function for grain mass and harvest index (Figure 4b,c) showed weak non-significant relationships with $CO_2$ concentration, with an $R^2$ of 0.15 and 0.087, respectively.

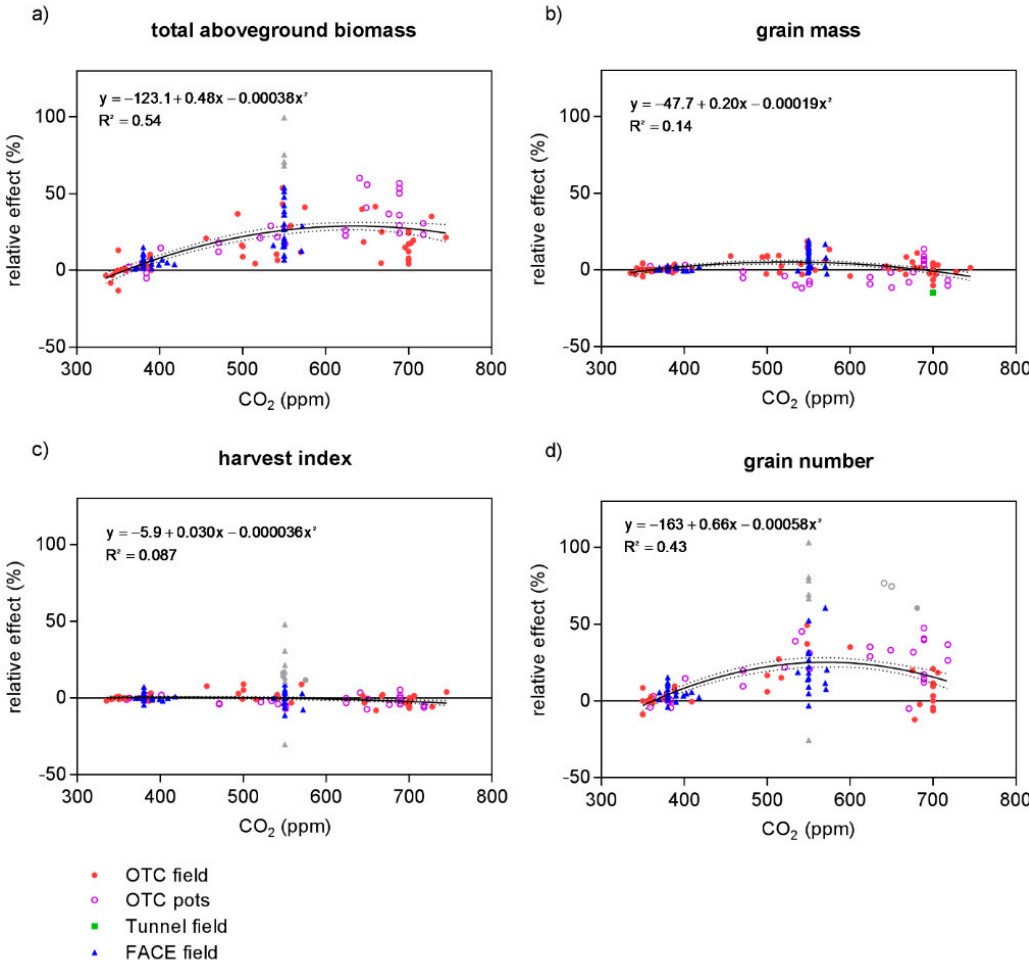

**Figure 4.** Response function for (**a**) total aboveground biomass, (**b**) grain mass, (**c**) harvest index, and (**d**) grain number with $CO_2$ concentration (relative to yield at 350 ppm). Dotted lines give the 95% confidence interval for the regression model. Grey markers are statistical outliers excluded from the curve fitting.

Based on the observation that the response pattern for the grain number was similar to grain yield, the correlation of these responses was tested. Figure 5 demonstrates the $eCO_2$ effect on grain number and the corresponding effect on grain yield, where the correlation of effects was very strong ($r = 0.82$). The broken line (1:1-line) represents the theoretical situation where the effect of $eCO_2$ on grain yield is entirely explained by the effect on grain number. The slope of the regression line ($y = 6.58 + 0.90x$) was not significantly different from the 1:1-line ($p = 0.20$).

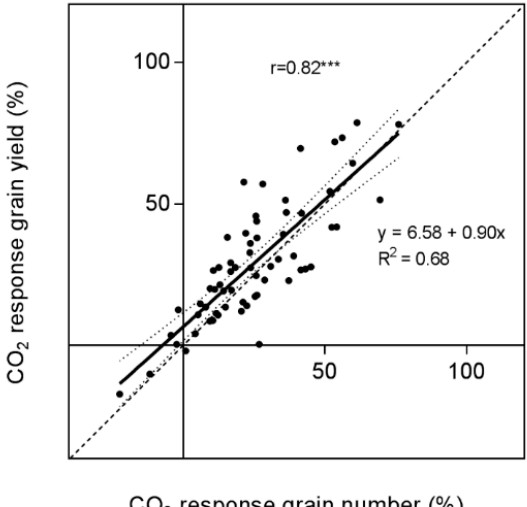

**Figure 5.** The $eCO_2$ effect on grain number versus the corresponding effect on grain yield. The broken line (1:1-line) represents the theoretical situation where the effect of $eCO_2$ on grain yield is entirely explained by the effect on grain number. Dotted lines give the 95% confidence interval for the regression model.

## 4. Discussion

### 4.1. Non-Linearity of $CO_2$ Response Function

From our analysis, we estimated that $eCO_2$, on average, increase wheat grain yield by 25% with an average $eCO_2$ level of 602 ppm. The gradual and non-linear $CO_2$ stimulation of grain yield found in this study agrees with the review of Amthor [19]. The best-fit response function for grain yield (Figure 2a) suggests a maximum grain yield stimulation around 600 ppm. Whereas the response function derived by Amthor [19] proposed that maximum yield response is reached at 890 ppm, but a close to linear yield increase in the range 350–750 ppm with slope of 0.086 for OTC and 0.080 for FACE, which is somewhat higher than our overall estimate of 0.062 (Table 1). Similar to this study, saturation of yield stimulation was also observed for potatoes grown under $eCO_2$ [29]. Previous meta-analyses used a beta factor to scale wheat yield response to the concentration difference between ambient and elevated $CO_2$ treatments [4,30,31], which assumes a linear response to rising $CO_2$ concentration. With the response functions derived in this study, we established that a linear scaling factor is not accurate for the $CO_2$ response of wheat grain yield and grain number. This is also supported by the observation of the non-existing relationship between grain yield $CO_2$ response and $\Delta CO_2$ (Figure 2b).

Our response function for grain yield suggests that there is a tendency for somewhat lower effects on yield at even higher $CO_2$ concentrations (above 600 ppm), though this is not possible to confirm with statistical significance based on the current dataset due to the lack of field experiments at very high $CO_2$ levels (none of the observations were above 750 ppm). There could hypothetically exist some negative impacts on plants grown under high $CO_2$ levels, which may offset the $CO_2$-induced increase in photosynthesis. Due to the lower transpiration rates, leaf canopy temperature can increase [32], which may cause heat stress and consequently earlier senescence, and thus impaired plant performance. Another consequence is a reduction in the transpiration-driven mass-flow that has been shown to be

closely linked to uptake of some plant nutrients (N, K, S, Ca, Mg, Mn) [33], which are essential for plant growth. At high $CO_2$ concentration, photosynthesis can be limited by triose-phosphate utilization [34] and $eCO_2$ may also impair N acquisition in leaves due to the suppressed photorespiration [35,36]. Since photorespiration drives the malate transport, $eCO_2$ is subject to impact the link between N and carbon metabolism in the plant. A lower photorespiration rate under $eCO_2$ results in a decreased supply of NADH, which powers the nitrate reduction, resulting in less N in leaves and grains. Plant demand for N is also higher since the total aboveground biomass increases under $eCO_2$. Consequently, growth and yield stimulation could be limited by N at very high $CO_2$, but, as already stated, the preset dataset was not sufficient to firmly assess the existence and magnitude of such an effect.

The response relationship for grain yield showed a certain degree of scatter, but the $CO_2$ concentration explains almost half of the variation in the data ($R^2 = 0.44$) when statistical outliers are excluded. In general, there is no systematic pattern of the data scatter, e.g., a similar variation in $eCO_2$ response for different types of exposure systems and rooting environments, with the exception of the fact that the four observations with strongest response (identified as statistical outliers) are all from FACE experiments. These observations originate from experiments conducted within the *Australian Grains Free Air $CO_2$ Enrichment* (AGFACE) project [37], located in a fairly dry and low productive agronomic environment where actual yields are considerably lower (average grain yield for Australian experiments is 243 g m$^{-2}$) than for the experiments performed in other regions (average grain yield for Asia, North America, and Europe is 650 g m$^{-2}$). It is, however, not surprising that the relative $eCO_2$ effect is stronger in dry environments due to the enhanced water use efficiency [4,37], and thus the extended duration of the grain filling period, where $CO_2$-induced water savings potentially can be an economical benefit for farmers irrigating their wheat fields.

### 4.2. Relation to Site Productivity

There is a clear link between grain yield response to $eCO_2$ and agronomic productivity with a higher relative yield response in regions of low productivity. In particular, no large $eCO_2$ grain yield stimulations were observed for observations with a high yield in the $aCO_2$ treatment. The relationship was independent of experimental systems (Figure 3a), level of $eCO_2$ treatment (Figure 3b), and geographical region (Figure 3c), and regression slopes (Table 2) were almost identical when comparing these subgroups, adding robustness to the conclusion that there is a general negative relationship between the grain yield $CO_2$ response and productivity of the site/year. Since this analysis was done only for a subset of data (experiments reporting grain yield reported in mass per area), it should, however, be noted that North America and Asia are not well represented and no separate regression analysis was performed for these continents (Figure 3c). From Figure 3a–c, it becomes obvious that experiments conducted in low productive systems (grain yield < 300 g m$^{-2}$) mainly use FACE systems and $eCO_2$ levels below 600 ppm, despite this, this is where we found the strongest yield responses. Comparing regression models fitted to the complete dataset (Figure 3a, Table 2) did not clearly show whether the relationship between yield $CO_2$ response and productivity was linear or not. In the range 200–900 g m$^{-2}$, regression lines were very similar, accordingly, both regression models are valid for the majority of wheat growing areas.

### 4.3. Experimental Conditions

A large fraction of additional data included in our analysis, as compared to the data used by Amthor [18], originated from FACE studies, which are often considered the most realistic experimental type, but this did not change the overall response pattern. From the meta-analysis (Figure 2c), we concluded that there were no significant differences in grain yield stimulation when comparing FACE experiments to those performed in OTCs. Thus, the influence of OTCs on microclimate, absent in FACE, was not of major importance for the $eCO_2$ effect on grain yield in wheat. Consequently, we can conclude that the results from OTC studies are still relevant and useful for the understanding of $eCO_2$ effects on wheat yield production. Our results consequently contrast with the findings by

Long et al. [31], who estimated yield stimulation to be about 50% less in FACE studies compared to enclosure studies. They did, however, not differentiate between open and closed-top chambers, which Wang, Feng, and Schjoerring [4] showed to differ significantly with respect to the $CO_2$ effect on grain yield response in wheat.

The overall comparison of $eCO_2$ effects on grain yield for wheat plants rooted in field soil and those grown in pots did not show any significant differences, and the same was true when only analyzing the subgroup of experiments using OTCs. Accordingly, the restricted rooting space for potted plants did not have a major influence on the $CO_2$ yield stimulation in wheat yield. This disagrees with the findings by Wang, Feng, and Schjoerring [4], where the grain yield response to $eCO_2$ in potted plants was significantly stronger than for plants rooted in field soil, but not taking exposure system into account. In contrast to the absence of a pot effect on the $eCO_2$ effect on grain yield in the present study, the $CO_2$-induced reduction of grain protein concentration was significantly stronger in potted plants compared to field grown ones [38,39]. Additionally, the difference in response has been shown to be related to pot size, where larger pots (>10 l) are comparable to field conditions [31]. Accordingly, the restricted rooting volume may be of greater importance for $eCO_2$ effects on nutrient uptake than for grain yield stimulation, especially when small pots are used.

### *4.4. Yield Components*

In addition to limitations of N and other nutrient sources, it has been argued that the $CO_2$ effect on growth can be dampened by sink limitations [40,41], such as by the maximum number of tillers and grains. Previous breeding improvements for wheat yield has mainly been a result of increase in harvest index [42,43], but from this study we can conclude that there was no significant $eCO_2$ effect on this yield factor (Figure 1). This is in agreement with previous studies where harvest index was reported to remain unaffected [37,44,45]. In contrast, Kimball et al. [46] observed a small but significant increase in harvest index by 4.4% in Arizona; discrepancies might be attributed to different environmental conditions (shorter vegetation period in Arizona, extra irrigation) and the cultivars used. If harvest index remains unaffected by $eCO_2$, the vegetative and reproductive parts of the plant are stimulated to the same extent. Thus, $eCO_2$ does not have an impact on redistribution of photosynthates from leaves and shoots to grains. Even though there still is a gap between the theoretical limit for harvest index (62%) and the actual harvest index achieved under field conditions [18], current breeding strategies are mainly focused on improving grain number [47], which is a yield component strongly and significantly enhanced by $eCO_2$ (Figure 1).

In some experiments, $eCO_2$ significantly reduced grain mass [31,44], whereas others showed a non-significant effect [37,48,49] or even a significant increase [45]. Despite this, Pleijel and Uddling [50] showed a weak but statistically significant relationship between the relative effects of $CO_2$ on grain yield and grain mass across a large number of experiments. Results from recent FACE experiments showed an increase in grain number under $eCO_2$ [44], which is in line with our results.

The $eCO_2$ wheat yield response may differ between cultivars due to the variable sink strength, which can be explored with screening strategies, e.g., in FACE facilities [51], both to identify more responsive cultivars, but also to better understand the link between genetics and traits related to high grain yield. If grain number is differently affected by $eCO_2$ in different wheat cultivars this could be considered in breeding programs, while, based on our investigation, it seems not to be a promising strategy to search variation in the $eCO_2$ response on grain mass and harvest index among existing cultivars.

### *4.5. Implications for Modelling and Food Security Assessment*

The response functions for grain yield and yield components are useful for application in crop modelling of wheat yield response to a gradual increase in $CO_2$ concentration, in the range up to 750 ppm. State-of-the-art models mainly assumes grain yield to continue to increase even at very high $CO_2$, in line with photosynthetic response to $eCO_2$ [8,52], which is in disagreement with our results.

Some existing wheat crop models accurately simulate grain yield and protein concentration under present-day environmental conditions, while representation of preceding growth dynamics is still challenging [53], which are of great importance to estimate impacts of environmental change.

It is crucial to consider interactions with other climate factors and crop management for better assessments of changes in crop yield production and food security under $eCO_2$. We have focused on $CO_2$ effects on yield and did not considered the interaction with other environmental factors, such as tropospheric ozone, temperature, soil fertility, and moisture. The positive effect of $eCO_2$ on crop yields is likely to be offset by ozone exposure together with heat and drought stress [54]. In contrast to the homogenous rise in $CO_2$, the impact from these stress factors will vary substantially both at a geographical and temporal scale, making projections for future crops yields rather complex. The $eCO_2$ stimulation of wheat yield is also dependent on N fertilization, where application rates up to 200 kg ha$^{-1}$ promotes the $eCO_2$ response [55].

When considering future food supply, both quantity and quality of crop yields are of great importance. We cannot simply focus on the amount of calories produced but also have to consider the potential impacts on nutritional quality. Even though $eCO_2$ is clearly stimulating wheat grain production, this benefit comes with adverse effects on food security. Previous studies [38,39,56,57] show that $eCO_2$ significantly reduced the concentrations of protein, iron, and zinc in wheat and other staple crops, including rice and maize, resulting in a lower nutritional quality. Myers et al. [58] estimated that 138 million people would be at risk of zinc deficiency by the year 2050, where the $CO_2$-induced reduction in zinc content of crops would significantly increase the number of people at risk. A similar assessment for protein found that by the year 2050, $eCO_2$ might cause an additional 148.4 million of the world's population to be at risk of protein deficiency [59].

## 5. Conclusions

In summary, from this updated research synthesis on $CO_2$ impacts on wheat crop yield, we conclude that grain yield stimulation does not respond linearly to increasing $CO_2$ but is likely to reach a maximum and level off already at ~600 ppm. Based on 92 observations in field experiments, yield is estimated to increase by 25% on average under $CO_2$ enrichment. This level of average response is independent of experimental facilities and geographic location. We attributed the $CO_2$-induced grain yield stimulation to an increase in total aboveground biomass and a larger number of grains, whereas harvest index and specific grain mass remain unaffected and grain mass only increases to a very small extent. Grain yield response also shows a strong link to site productivity, where relative response is larger in low productive systems.

**Supplementary Materials:** The following are available online at http://www.mdpi.com/2073-4395/9/5/243/s1, Supplementary file: Supplementary Database.

**Author Contributions:** M.C.B. and H.P. conceptualized and designed the study; data collection was made by M.C.B. in collaboration with P.H. and Z.F., M.C.B. performed data analysis and original draft preparation. Writing and editing was done by M.C.B., P.H., Z.F., and H.P. collaboratively.

**Funding:** The strategic research area BECC (Biodiversity and Ecosystem Services in a Changing Climate; http://www.cec.lu.se/research/becc) supported this work, and it was performed partly within the bilateral Sweden–China framework research program on "Photochemical smog in China: formation, transformation, impact and abatement strategies", supported by the Swedish Research Council (639-2013-6917). This work was also granted by the German Research Foundation as part of the integrated project "Structure and Functions of Agricultural Landscapes under Global Climate Change–Processes and Projections on a Regional Scale" (PAK 346) and the Research Unit FOR 1695 "Agricultural Landscapes under Global Climate Change–Processes and Feedbacks on a Regional Scale" (https://klimawandel.uni-hohenheim.de/home).

**Conflicts of Interest:** The authors declare no conflict of interest.

# Appendix A

**Table A1.** References for data sources used in the analysis, database in the Supplementary Materials file.

| Reference | Grain Yield | Total Aboveground Biomass | Harvest Index | Grain Mass | Grain Number | Specific Grain Mass |
|---|---|---|---|---|---|---|
| Deepak and Agrawal, 1999 [49] | x | | | x | x | |
| Dietterich et al., 2015 [60] | x | x | x | x | x | |
| Dijkstra et al., 1999 [61] | x | x | x | x | x | |
| Donnelly et al., 1999 [62] | x | | | x | x | |
| Fangmeier et al., 1996 [63] | x | x | x | x | x | |
| Hakala, 1998 [64] | x | x | x | x | x | |
| Houshmandfar et al., 2016 [65] | x | x | x | x | x | |
| Högy, 2002 [66] | x | x | x | x | x | |
| Högy et al., 2009 [44]* | x | x | x | x | x | |
| Högy et al., 2013 [45] | x | x | x | x | x | |
| Kimball et al., 2001 [48] | x | x | x | x | | x |
| Manderscheid and Weigel, 1997 [67] | x | x | x | x | x | |
| Mishra et al., 2013 [68] | x | x | x | x | x | |
| Mortenssen et al. [69]; Fangmeier et al., 1999 [70] | x | x | x | x | | |
| Mulholland et al., 1998 [71] | x | x | x | x | | |
| Fangmeier et al., 1999 [70] Piikki et al., 2008 [72]; | x | x | x | x | | x |
| Pleijel et al., 2000 [73] | x | x | x | x | | |
| Rawson 1995 [74] | x | | | x | x | |
| Mulchi et al., 1995 [75]; Rudorff et al., 1996a [76]; Rudorff et al 1996b [77] | x | x | x | x | x | x |
| Van Oijen et al., 1999 [78] | x | x | x | x | x | |
| Weigel and Manderscheid 2012 [79] | x | x | x | | | |
| Manderscheid et al., 1995 [80]; Weigel et al., 1994 [81] | x | x | x | x | x | |
| Yang et al., 2007a [82]; Yang et al., 2007b [83] | x | | | x | x | |

\* Data for individual years obtained directly from the authors.

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
