# Peer review of "Effects of Elevated CO2 on Wheat Yield: Non-Linear Response and Relation to Site Productivity"

_agronomy, doi:10.3390/agronomy9050243_

Round 1
Reviewer 1 Report
By means of a meta-analysis of published data, Malin et al study how field-grown wheat growth and yield is increased under different concentrations of eCO2. Their main finding is that the response to eCO2 is nonlinear, i.e. growth and yield stimulation plateaus at about 600 ppm. Growth stimulation is also stronger with decreasing regional productivity.
It is the first time I review this manuscript. The meta-analysis is conducted adequately and, for as far as I can judge, the literature is well covered. The point of a non-linear response is useful for the field. Although perhaps not very striking, I think this paper was well written. Also the comparison between different treatment types (e.g. FACE, OTC) is useful.
I have no major changes to insist upon, only very few specific points.
L83-95: Perhaps it would be pertinent to here also mention another recent overview study that included wheat (Environmental and Experimental Botany (2016) 122:150-157).
L139: meta-analysis à analyses
One final conceptual point I would like to raise is the fact that the authors deduce significance from non-overlapping confidence intervals (see paragraph 2.2). In many cases the differences found in this paper are so obvious that no statistics are needed; however, in case of the between geographic region comparison, where the authors discuss some possible differences and (limited) sample sizes, there may well be no statistical difference when tested with proper statistics. For example a permutation approach would work, but probably also meta-analysis specific tests will do.
Author Response
Response is found in the attached word-file

Reviewer 2 Report
The paper presents a meta-analysis of an updated number of experimental studies consisting in comparing wheat yield, grain number, grain weight and specific grain weight under ambient and elevated CO2 concentration (eCO2). The authors show that the relationship between yield and CO2 concentration is not linear and that no increase in yield occurs beyond a concentration of 600 ppm approximately. The maximum increase in yield, i.e. between ambient CO2 and 600 ppm was approximately 26 % on average. Grain number increased as much as yield in under eCO2 conditions. The CO2 impact did not depend on cultivar, experimental design (FACE, OTC or tunnel) or whether plants were cultivated in pots or in the field (OTC). Finally a clear negative correlation appeared between the CO2 impact and the control yield, showing that CO2 showed its largest impact where fertility (mainly ascribed to local water availability) was lowest.
The paper is generally clearly written and convincing. It confirms some earlier results but with more and recent results, bringing about a higher precision in the general assessment of CO2 impact. This is an important contribution to the current debate on the impact of climate change on future wheat yields, one of the 3 major cereals in the world.
I only have small remarks on the manuscript.
Title: Replace the “-” by “:”
Abstract: The mean range of [CO2] corresponding to the 26 % increase in yield should be given (370/600 ?)
line 47-49. I don’t know whether it is good to state those hypothesis, which turn out not to be confirmed by the observations. The study has no data on leaf physiology or gas exchange. Maybe better to drop this.
line 98. The only way to assess the optimality of N supply is to check that the nitrogen status of the crop is non limiting (eg: nitrogen concentration of the biomass referred to a critical concentration depending on standing biomass). However there is no information about such N status in the dataset. As elevated CO2 is known to bring about a dilution of N in the plant biomass, a certain level of N stress might be expected if no correction is applied for N fertilization. Also this might be a part of the explanation of the negative relationship between biomass and CO2.
Line 124. What are those data “obtained directly from experimenters” ?. Not presented experiments ? If some data are used, which were not published earlier, they should be described in the material and method section. I am not sure that such dataset can be included in a meta-analysis.
Fig 2 a: I get grey points on the graph I printed from the Agronomy site. Is it a bug in the PDF ? Or is there more than what is depicted in in the legend ?
Line 282. Maybe the Amthor 2009 line could be shown on the Fig 2a.
Line 304. As N fertilization is not changed in eCO2 versus control treatments, N might become limiting at eCO2. This is an alternative explanation to be discussed.
line 324. The dependance of the CO2 impact on local control wheat yield at ambient CO2 does not seem consistent with findings by O’Leary et al. (2015) in the Australian Grain FACE (O'Leary, G. J., Christy, B., Nuttall, J., Huth, N., Cammarano, D., Stöckle, C., ... & Farre‐Codina, I. (2015). Response of wheat growth, grain yield and water use to elevated CO 2 under a free‐air CO 2 enrichment (FACE) experiment and modelling in a semi‐arid environment. Global change biology, 21(7), 2670-2686.) . The discussion should mention that publication in that discussion section.
Line 403-404. That statement seems in contradiction with the fact that N fertilization was optimal.
Author Response

(The authors gave the same response as above.)
